# Synthesis of molecular metallic barium superhydride: pseudocubic BaH$_{12}$

Wuhao Chen[1,8], Dmitrii V. Semenok [2,8], Alexander G. Kvashnin [2,8], Xiaoli Huang [1✉], Ivan A. Kruglov[3,4], Michele Galasso [2], Hao Song[1], Defang Duan [1], Alexander F. Goncharov [5], Vitali B. Prakapenka [6], Artem R. Oganov [2✉] & Tian Cui[1,7✉]

Following the discovery of high-temperature superconductivity in the La–H system, we studied the formation of new chemical compounds in the barium-hydrogen system at pressures from 75 to 173 GPa. Using in situ generation of hydrogen from NH$_3$BH$_3$, we synthesized previously unknown superhydride BaH$_{12}$ with a pseudocubic (*fcc*) Ba sublattice in four independent experiments. Density functional theory calculations indicate close agreement between the theoretical and experimental equations of state. In addition, we identified previously known *P6/mmm*-BaH$_2$ and possibly BaH$_{10}$ and BaH$_6$ as impurities in the samples. Ab initio calculations show that newly discovered semimetallic BaH$_{12}$ contains H$_2$ and H$_3^-$ molecular units and detached H$_{12}$ chains which are formed as a result of a Peierls-type distortion of the cubic cage structure. Barium dodecahydride is a unique molecular hydride with metallic conductivity that demonstrates the superconducting transition around 20 K at 140 GPa.

[1] State Key Laboratory of Superhard Materials, College of Physics, Jilin University, Changchun 130012, China. [2] Skolkovo Institute of Science and Technology, Skolkovo Innovation Center, 3 Nobel Street, Moscow 143026, Russia. [3] Moscow Institute of Physics and Technology, 9 Institutsky Lane, Dolgoprudny 141700, Russia. [4] Dukhov Research Institute of Automatics (VNIIA), Moscow 127055, Russia. [5] Earth and Planets Laboratory, Carnegie Institution of Washington, 5251 Broad Branch Road NW, Washington, DC 20015, USA. [6] Center for Advanced Radiation Sources, The University of Chicago, 5640 South Ellis Avenue, Chicago, IL 60637, USA. [7] School of Physical Science and Technology, Ningbo University, Ningbo 315211, China. [8] These authors contributed equally: Wuhao Chen, Dmitrii V. Semenok, Alexander G. Kvashnin. ✉email: huangxiaoli@jlu.edu.cn; a.oganov@skoltech.ru; cuitian@jlu.edu.cn

In recent years, the search for new hydride superconductors with $T_C$ close to room temperature attracts great attention of researchers in the field of high-pressure materials science. Variation of pressure opens prospects of synthesis of novel functional materials with unexpected properties[1]. For example, according to theoretical models[2–5], compression of molecular hydrogen over 500 GPa should lead to the formation of an atomic metallic modification with $T_C$ near room temperature. Pressures of 420–480 GPa were achieved in experiments with toroidal diamond anvil cells[6]; however, for conventional high-pressure cells with a four-electrode electric setup, pressures above 200 GPa remain challenging.

In 2004, Ashcroft[7] suggested an alternative method of searching for high-$T_C$ superconductors that uses other elements, metals or nonmetals, to precompress the hydrogen atoms, which should lead to a dramatic decrease in the metallization pressure. A decade later this idea found its experimental proof. Extraordinarily high superconducting transition temperatures were demonstrated in compressed $Im\bar{3}m$-$H_3S$[8–11] (203 K at 150 GPa), $Im\bar{3}m$-$YH_6$[12] and $P6_3/mmc$-$YH_9$[13] (224 K at 166 GPa and 243 K at 237 GPa, respectively), $Fm\bar{3}m$-$ThH_{10}$[14] (161 K at 174 GPa), $P6_3/mmc$-$CeH_9$[15] (~100 K), and lanthanum decahydride $Fm\bar{3}m$-$LaH_{10}$[16–18] with $T_C > 250$ K at 175 GPa.

The neighbor of lanthanum, barium is a promising element for superhydride synthesis. The calculated maximum $T_C$ is only about 30–38 K[19,20] for predicted $P4/mmm$-$BaH_6$ stable at 100 GPa, which has a hydrogen sublattice consisting of $H_2$ molecules and $H^-$ anions[19]. Lower barium hydride, $BaH_2$, well-known for its extraordinarily anionic ($H^-$) conductivity[21], exists in $Pnma$ modification below 2.3 GPa, whereas above 2.3 GPa it undergoes a transition to hexagonal $Ni_2In$-type $P6_3/mmc$ phase[22]. At pressures above 41 GPa, $BaH_2$ transforms into $P6/mmm$ modification, which metallizes at over 44 GPa, but its superconducting $T_C$ is close to zero[23]. So far, no relevant experiments at pressures above 50 GPa have been reported.

In this work we experimentally and theoretically investigate the chemistry of the barium-hydrogen system at pressures from 75 to 173 GPa filling the gap of previous studies. We discover new pseudocubic $BaH_{12}$ that has molecular structure with $H_2$ and $H_3^-$ molecular units and detached $H_{12}$ chains formed due to Peierls-type distortion. These structural features lead to metallic conductivity of unique molecular hydride and to the superconducting transition around 20 K at 140 GPa.

## Results

**Synthesis at 160 GPa and Stability of $BaH_{12}$.** To investigate the formation of new chemical compounds in the Ba–H system at high pressures, we loaded four high-pressure diamond anvil cells (DACs #B0-B3) with sublimated ammonia borane $NH_3BH_3$ (AB), used as both a source of hydrogen and a pressure transmitting medium. A tungsten foil with a thickness of about 20 μm was used as a gasket. Additional parameters of the high-pressure diamond anvil cells are given in Supplementary Table S1.

The first attempt of the experimental synthesis was made in DAC #B1 heated to 1700 K by an infrared laser pulse with a duration of ~0.5 s at a pressure of 160 GPa. During heating, the Ba particle underwent significant expansion and remained nontransparent. The obtained synchrotron X-ray diffraction pattern (XRD, $\lambda = 0.62$ Å, Fig. 1a) consists of a series of strong reflections specific to cubic crystals. Decreasing the pressure in DAC #B1 to 119 GPa (Fig. 1b) gave a series of diffraction patterns that can mostly be indexed by a slightly distorted face-centered cubic structure (e.g., pseudocubic $Cmc2_1$, Fig. 1a). Recently, similar cubic diffraction patterns have been observed at pressures above 150 GPa for the La–H ($fcc$-$LaH_{10}$)[17,18] and Th-H ($fcc$-$ThH_{10}$)[14] systems. By analogy with the La–H system, and considering the lack of previously predicted cubic superhydrides $BaH_x$[19–21], we used the USPEX code[24–26] to perform theoretical crystal structure evolutionary searches, both variable- and fixed-composition, for stable Ba–H compounds at pressures of 100–200 GPa and temperatures of 0–2000 K.

According to the USPEX calculations, $P6/mmm$-$BaH_2$ remains stable up to 150–200 GPa (Fig. 1c; Supplementary Tables S7–S12, Supplementary Figs. S2 and S3). This compound was experimentally detected in DAC #B0 at 173–130 GPa with the cell volume ~3% smaller than theoretically predicted (Supplementary Table S14). At 100–200 GPa, several new barium polyhydrides lying on or near the convex hulls were found: $BaH_6$, $BaH_{10}$, and $BaH_{12}$ with the unit cell $Ba_4H_{48}$ and $Ba_8H_{96}$ (Fig. 1c). In subsequent experiments at 142 and 154–173 GPa we have detected a series of reflections that can be indexed by $BaH_6$ and $BaH_{10}$ with the unit cell volumes close to the calculated ones (see Supporting Information, p. S25-27). However, the main phase in almost all diffraction patterns is the pseudocubic barium superhydride which will be described below.

The analysis of the experimental data within space group $Fm\bar{3}m$ (Fig. 1b and Supplementary Table S3) of Ba-sublattice and its comparison with density functional theory (DFT) calculations show that the stoichiometry of barium hydride synthesized in DAC #B1 is close to $BaH_{12}$. Examining the results of the fixed-composition search, we found that an ideal $Fm\bar{3}m$-$BaH_{12}$ (similar to $fcc$-$YB_{12}$) is unstable and cannot exist, while pseudocubic $P2_1$-$BaH_{12}$, whose predicted diffraction pattern is similar to the experimental one, lies on the convex hull at 100–150 GPa. There are also pseudocubic $P1$-$Ba_8H_{96}$, located very close to the convex hull at 150 GPa, and $Cmc2_1$-$BaH_{12}$ (= $Ba_4H_{48}$) with a similar X-ray diffraction (XRD) pattern, lying a bit farther. Above 190 GPa the $P2_1$-$BaH_{12}$ transforms to other possible candidate, orthorhombic $Immm$-$BaH_{12}$, which stabilizes between 150 and 200 GPa, but does not correspond to the experimental XRD pattern (Fig. 1a, Supplementary Fig. S1) and is not considered further.

The computed equation of state of $Fm\bar{3}m$-$BaH_{12}$ (Fig. 1d) corresponds well to the experimental volume-pressure dependence above 100 GPa. However, the DFT calculations show that the ideal $Fm\bar{3}m$ barium sublattice is unstable (it is $> 0.19$ eV/atom above the convex hull, Supplementary Fig. S4) both thermodynamically and dynamically, and transforms spontaneously to $Cmc2_1$ or $P2_1$ via distortion (Fig. 2). Studying the temperature dependence of the Gibbs free energy (Fig. 2a), we found that $P2_1$-$BaH_{12}$ is the most stable modification at 0–2000 K and 100–150 GPa. Moreover, high-symmetry cubic phases cannot explain the weak reflections at 8.9–9.4°, 14.5, 16, 19.5, and 20.6° present in many XRD patterns (Fig. 1a, b).

To clarify the question of dynamical stability of pseudocubic structures, we calculated a series of phonon densities of states for different modifications of $BaH_{12}$ (Fig. 2b). Within the harmonic approach, symmetric and corresponding to the experimental data $Cmc2_1$-$BaH_{12}$ has a number of imaginary phonon modes. Its distortion to much more stable $P2_1$-$BaH_{12}$ leads to the disappearance of many of the imaginary phonon modes and deepening of the pseudogap (Fig. 2c) in the electronic density of states $N(E)$. The subsequent distortion of $P2_1$ to $P1$ converts $BaH_{12}$ to a semiconductor with a bandgap exceeding 0.5 eV. However, the experimental data show that $BaH_{12}$ remains opaque in the visible range, does not give Raman signals (Supplementary Figs. S39-40), retains an almost $fcc$ crystal structure, and exhibits metallic properties (see next sections) down to 75 GPa. For this reason, the electronic band structure and parameters of the superconducting state were further investigated only for $Cmc2_1$-$BaH_{12}$, which does not have a bandgap at 100–150 GPa. Stability

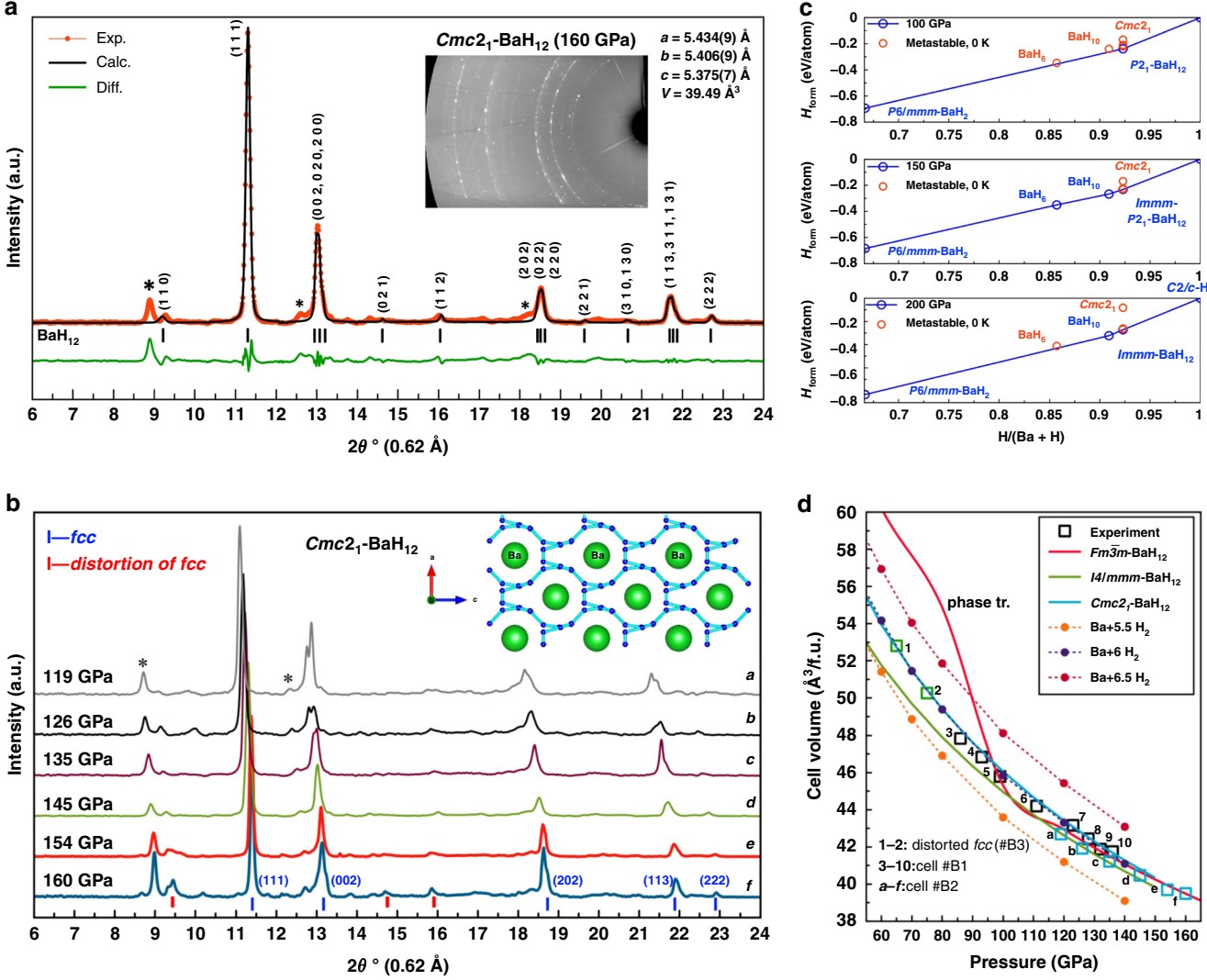

**Fig. 1 XRD patterns of synthesized samples at various pressures with theoretical analysis of their stability. a** Experimental X-ray diffraction pattern from DAC #B1 at 160 GPa and the Le Bail refinement of the pseudocubic $Cmc2_1$-BaH$_{12}$ phase. The experimental data, fitted line, and residues are shown in red, black, and green, respectively. Unidentified reflections are indicated by asterisks. **b** X-ray diffraction patterns at pressures of 119 to 160 GPa. The inset shows the projection of the $Cmc2_1$ structure to the ($ac$) plane. The hydrogen network is shown by light blue lines. **c** Convex hulls of the Ba–H system at 100, 150, and 200 GPa calculated with zero-point energy (ZPE) contribution. **d** Calculated equations of state for different possible crystal modifications of BaH$_{12}$ ($fcc$, $I4/mmm$, and $Cmc2_1$) and Ba+$n$H$_2$. The experimental data are shown by hollow squares.

of all considered polymorphic modifications of BaH$_{12}$ at different pressures with respect to other Ba-H phases and with respect to each other are shown in Supplementary Figs. S4 and S5 (see Supporting Information).

The comparative analysis of $Cmc2_1$, $P2_1$, and $P1$ structures of BaH$_{12}$ shows that semimetallic $Cmc2_1$ explains well the experimental results of X-ray diffraction (see Supplementary Fig. S1, Supporting Information) and lies closer to the convex hull than $Fm\bar{3}m$ or $I4/mmm$ modifications. $P1$-BaH$_{12}$ shows a complex picture of splitting of the diffraction signals, both $P1$-BaH$_{12}$ and $P2_1$-BaH$_{12}$ have a bandgap above 0.5 eV at 100 GPa (Supplementary Fig. S38b) which does not correspond to the experimental data. Therefore, pseudocubic $Cmc2_1$-BaH$_{12}$, whose cell volume is near that of the close-packed $Fm\bar{3}m$-BaH$_{12}$, is the appropriate explanation of the experimental results despite the presence of a few imaginary phonon modes.

The molecular dynamics simulation of $Cmc2_1$-BaH$_{12}$ and $P2_1$-BaH$_{12}$ at 10–1500 K, after averaging the coordinates, both lead to a distorted pseudocubic $P1$-BaH$_{12}$ with the similar XRD pattern.

However, all structures retrieved by molecular dynamics are less stable both dynamically and thermodynamically than $P1$-BaH$_{12}$, $P2_1$-BaH$_{12}$, and $Cmc2_1$-BaH$_{12}$ found by USPEX. More accurate analysis accounting for the anharmonic nature of hydrogen oscillations[27], which is actually beyond the scope of this work, may help to explain the experimental stability of higher-symmetry BaH$_{12}$ modifications compared to lower-symmetry $P1$-BaH$_{12}$.

**Synthesis of BaH$_{12}$ at 146 GPa.** Similar X-ray diffraction patterns were obtained in the next experiment (DAC #B2) where the Ba sample was heated at an initial pressure of 146 GPa, which led to a decrease in pressure to 140 GPa. During the heating and subsequent unloading of the cell, the sample remained opaque down to ~40 GPa. Unlike the synthesis at high pressure (cell #B1, 160 GPa, Fig. 1a, b), in this experiment we observed many more side phases and corresponding side reflections than before (Fig. 3 and Supporting Information).

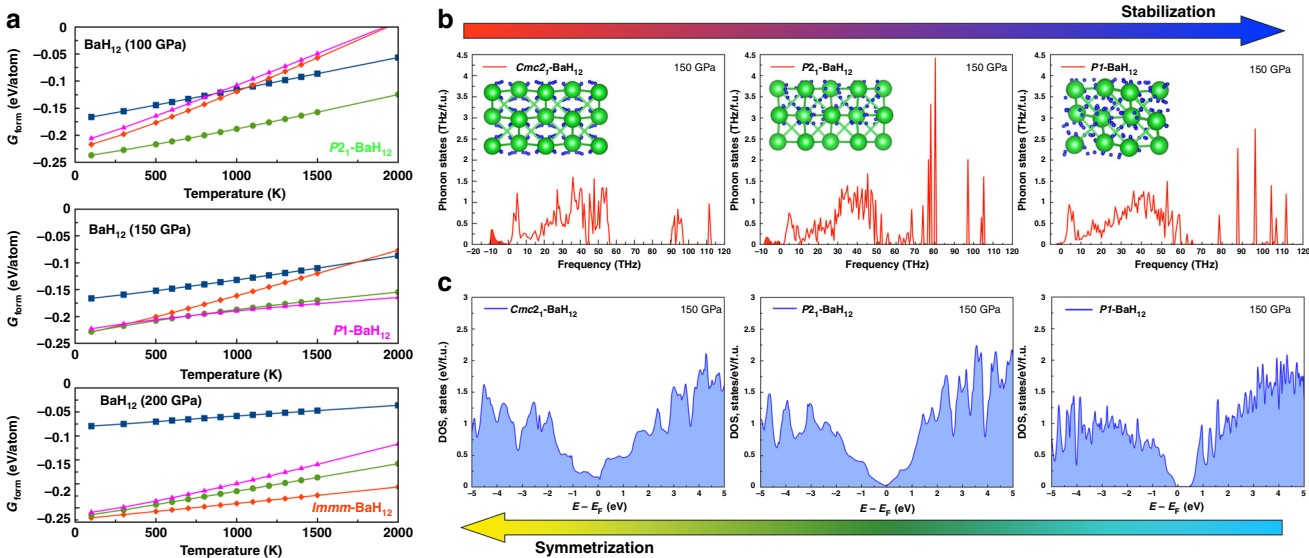

**Fig. 2 Theoretical study of stability and electronic properties. a** Dependence of the Gibbs free energies of formation on the temperature for different modifications of BaH$_{12}$ — *Immm*, pseudocubic *Cmc*2$_1$, *P*2$_1$, and *P*1 — calculated within the harmonic approach in the temperature range of 100 to 2000 K at 100, 150, and 200 GPa. **b** Phonon and **c** electron densities of states for pseudocubic BaH$_{12}$ structures with various degrees of distortion: *Cmc*2$_1$, *P*2$_1$, and *P*1.

Similar to the experiment with DAC #B1, five reflections from the pseudocubic Ba sublattice dominate in a wide range of pressures (65–140 GPa), whereas side reflections change their intensities and, at some pressures, almost disappear (Fig. 3d and Supplementary Figs. S33 and S35). The diffraction circles corresponding to the ideal cubic barium sublattice have pronounced granularity (Fig. 3e–g, Supplementary Fig. S34), which suggests that all "cubic" reflections belong to the same phase.

At pressures below 65 GPa, it is no longer possible to refine the cell parameters of pseudocubic BaH$_{12}$. The parameters of the *Cmc*2$_1$-BaH$_{12}$ unit cell, refined to the experimental data, are presented in Supplementary Table S6. Fitting this pressure-volume data in the pressure range from 75 to 173 GPa by the third-order Birch–Murnaghan equation of state[28] gives the cell volume $V_{100} = 45.47 \pm 0.13$ Å$^3$, bulk modulus $K_{100} = 305 \pm 8.5$ GPa, and its derivative with respect to pressure $K'_{100} = 3.8 \pm 0.48$ (the index 100 designates values at 100 GPa). Fitting the theoretical data yields similar values: $V_{100} = 46.0$ Å$^3$, $K_{100} = 315.9$ GPa, and $K'_{100} = 2.94$.

**Synthesis of BaH$_{12}$ at 90 GPa.** In the experiment with DAC #B3, we investigated the possibility to synthesize BaH$_{12}$ at pressures below 100 GPa. After the laser heating of Ba/AB to 1600 K, the pressure in the cell decreased from 90 to 84 GPa. The observed diffraction pattern is generally similar to those in the previous experiments with DAC #B1, except the presence of the impurity, $h$-BaH$_{\sim 12}$, whose reflections may be indexed by hexagonal space groups $P6_3/mmc$ or $P6_3mc$ ($a = 3.955(7)$ Å, $c = 7.650(7)$ Å, $V = 51.84$ Å$^3$ at 78 GPa). For the main set of reflections, slightly distorted cubic BaH$_{12}$ is the best solution (Fig. 4). The refined cell parameters of BaH$_{12}$ (Supplementary Table S4) agree well with the results obtained previously with DACs #B1 and B2. When the pressure was reduced to 78 GPa, barium dodecahydride began to decompose, and subsequent diffraction patterns (e.g., at 68 GPa, see Supporting Information) show a complex image of broad reflections that confirms the lower experimental bound of BaH$_{12}$ stability of ~75 GPa mentioned above.

## Discussion

**Electronic properties of BaH$_{12}$.** BaH$_{12}$ is the first known metal hydride with such a high hydrogen content that is stable at such low pressures (~75 GPa). We further investigated its electronic structure and the charge state of the hydrogen and barium atoms. The electron localization function (ELF) analysis[29] (Fig. 4e–g) shows that hydrogen in BaH$_{12}$, similar to NaH$_7$[30], is present in the form of H$_2$ ($d_{\text{H–H}} = 0.78$ Å) and almost linear H$_3$ ($d_{\text{H–H}} = 0.81$ and $1.07$ Å) molecular fragments that form separate flat horseshoe-like H$_{12}$ chains ($d_{\text{H–H}} < 1.27$ Å, Fig. 4).

Bader charge analysis of *Cmc*2$_1$-BaH$_{12}$, performed in accordance with our previous experience[31,32] (Supplementary Table S18), shows that the Ba atoms serve as a source of electrons for the hydrogen sublattice. The charge of the barium atoms in BaH$_{12}$ is +1.15 at 150 GPa, whereas most of the hydrogen atoms have a negative charge. In the H$_3$ fragments, the charge of the end atoms is close to –0.2 and –0.27, while the H bridge has a small positive charge of +0.06 (Fig. 4e–g). In general, H$_3^-$ anion, similar to one found in the structure of NaH$_7$[30], has a total charge of –0.4 |$e$|, whereas molecular fragments H$_2$ ($d_{\text{H–H}} = 0.78$ Å) have a charge of only –0.1 |$e$|. Therefore, the Ba–H bonds in BaH$_{12}$ have substantial ionic character, whereas the H–H bonds are mainly covalent.

The low electronic density of states $N(E)$ in semimetallic *Cmc*2$_1$-BaH$_{12}$ looks typical for one-dimensional …H–H–H… chains (Fig. 4h, Supplementary Figs. S36 and S38) which are divided into H$_2$, H$_3$ fragments due to the Peierls-type distortion[33]. In fact, all of the discussed structures of BaH$_{12}$ can be viewed as a result of Peierls-type distortion. The main contribution to $N(E_{\text{F}})$, 83% at 150 GPa, comes from hydrogen (Fig. 4h), and ¾ of this contribution is related to s orbitals. At 150 GPa, barium in BaH$_{12}$ exhibits the properties of a d-block element, and its bonding orbitals have a significant d-character (Fig. 4i). Electrical conductivity is localized in the H layers consisting of quasi-one-dimensional …H–H–H… chains which are interconnected in non-trivial way (Fig. 4e–g, Supplementary Table S2 for crystal structure). Thus, barium dodecahydride is the first known molecular superhydride with metallic conductivity embedded in layers and one-dimensional chains of molecular hydrogen.

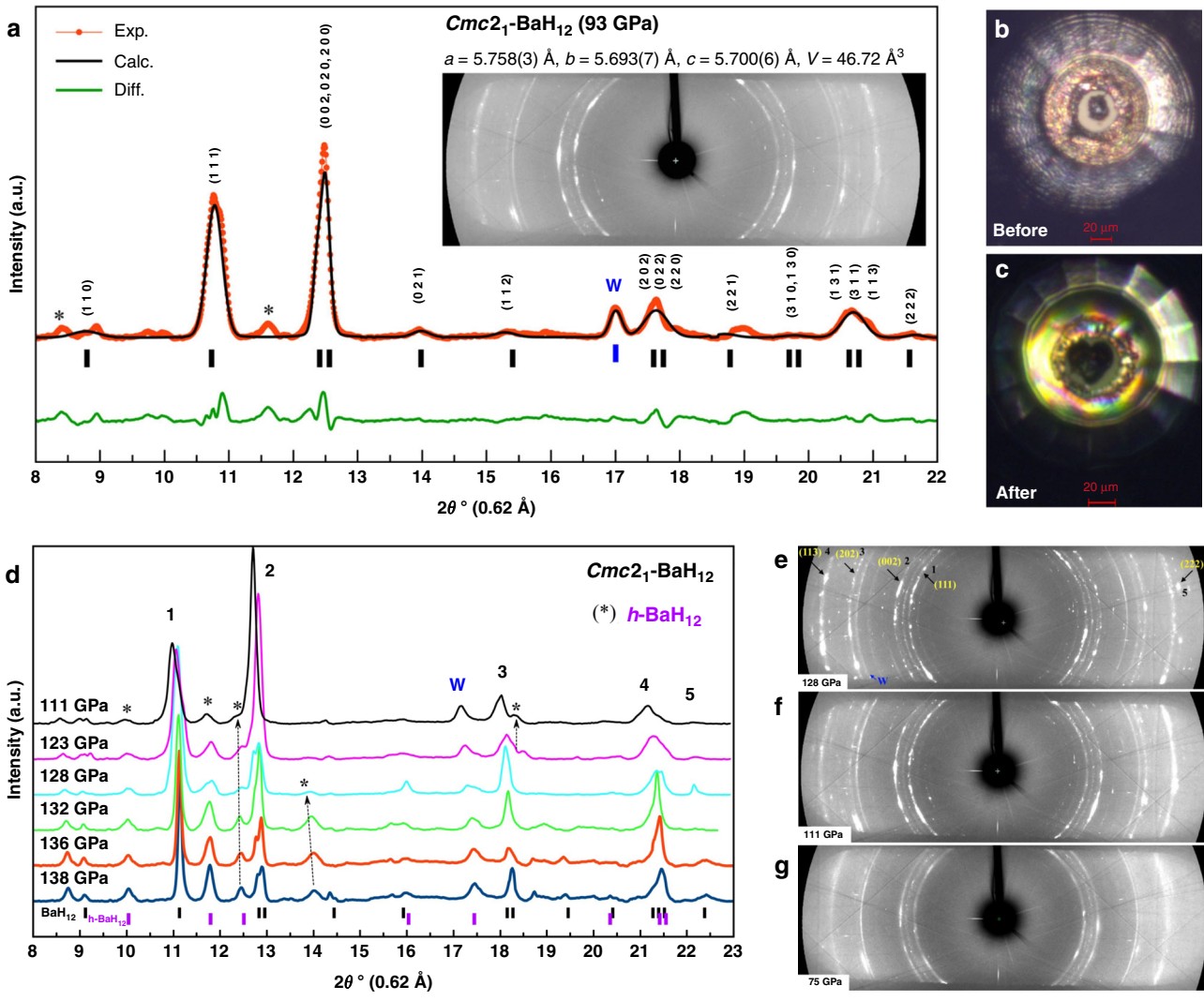

**Fig. 3 Analysis of experimental XRD patterns of *Cmc*2$_1$-BaH$_{12}$ synthesized in DAC #B2. a** Le Bail refinement of pseudocubic *Cmc*2$_1$-BaH$_{12}$ and *bcc*-W at 93 GPa. The experimental data, fitted line, and residues are shown in red, black, and green, respectively. Inset shows the 2D diffraction image. Weak reflections, indicated by asterisks, may correspond to the impurity: possibly *P*6$_3$/*mmc* or *P*6$_3$*mc*-BaH$_{12}$. **b, c** Microphotographs of the culet of cell #B2 with the Ba/AB sample before and after the laser heating. **d** Experimental XRD patterns from cell #B2 at pressures decreasing from 138 to 111 GPa. **e, f, g** Powder X-ray diffraction images at 128, 111, and 75 GPa. Reflections corresponding to the crystallographic planes in the ideal *Fm*$\bar{3}$*m*-BaH$_{12}$ phase are indicated by arrows.

**Superconductivity of BaH$_{12}$.** On the basis of powder diffraction experiments and thermodynamic calculations alone, we cannot unambiguously determine the structure of the H sublattice in BaH$_{12}$, which is essential for understanding superconductivity. To clarify this, we measured the electrical resistance $R$ of barium hydride samples using the well-known four-probe technique in the temperature range of 2–300 K. At pressures of 90–140 GPa, all five BaH$_x$ samples (DACs E#1-5, see Supporting Information) synthesized in electric DACs behave as typical metals with an almost linear decrease of $R(T)$. At low temperatures the resistance of the samples drops sharply, indicating a possible superconducting transition at about 5–7 K below 130 GPa (Supplementary Fig. S41), and ~20 K at 140 GPa (cell #E5, Fig. 5a). This DAC #E5 was assembled with an 80 μm diamond anvil culet, *c*-BN/epoxy insulating gasket, 45 × 32 μm Ba piece, and sputtered 0.5 μm thick Mo electrodes. After the laser heating at 1600 K and 140 GPa, the Ba/AB sample demonstrated the superconducting transition at around 20 K (Fig. 5a). When we tried to change the pressure, the cell collapsed and pressure dropped to 65 GPa.

The obtained data together with the measured Raman spectra and optical microscopy exclude low-symmetry BaH$_{12}$

semiconducting structures, leaving for consideration only metallic and semimetallic modifications (Supplementary Figs. S39 and S40).

The harmonic DFT calculations (Fig. 5b) demonstrate that the low density of electronic states near the Fermi level in *Cmc*2$_1$-BaH$_{12}$ is associated with a weak electron-phonon coupling, mostly related to low-frequency Ba and H phonon modes (1–10 THz), resulting in relatively low $\lambda = 1.02$, $\omega_{log} = 677$ K, $T_C = 39$–53 K, and $\mu_0 H_{c2}(0) = 5.1$–7 T at 150 GPa ($\mu^* = 0.15$–0.1). Decreasing pressure leads to a decrease of $\lambda$ and $T_C$ (from 53 to 46 K) at 140 GPa with a slope $dT_C/dP = 0.7$ K/GPa.

One of the roles of metal atoms in superhydrides is to donate electrons to antibonding orbitals of the H$_2$ molecules and weaken the H-H bonds. In BaH$_{12}$, each H atom accepts few electrons, on average 0.16 electrons. As a result, H$_2$ and H$_3$ groups are still present in the structure, and we have a rather low $T_C$. We think that at high pressures, due to dissociation of molecular groups, BaH$_{12}$ may have a network of weak H-H bonds (rather than discrete H$_2$ and H$_3$-groups) and, as a result, a much higher $T_C$. Increasing the pressure will also facilitate further metallization of BaH$_{12}$ and symmetrization of the hydrogen sublattice, increasing $N(E_F)$. To estimate the possible improvement, we calculated at

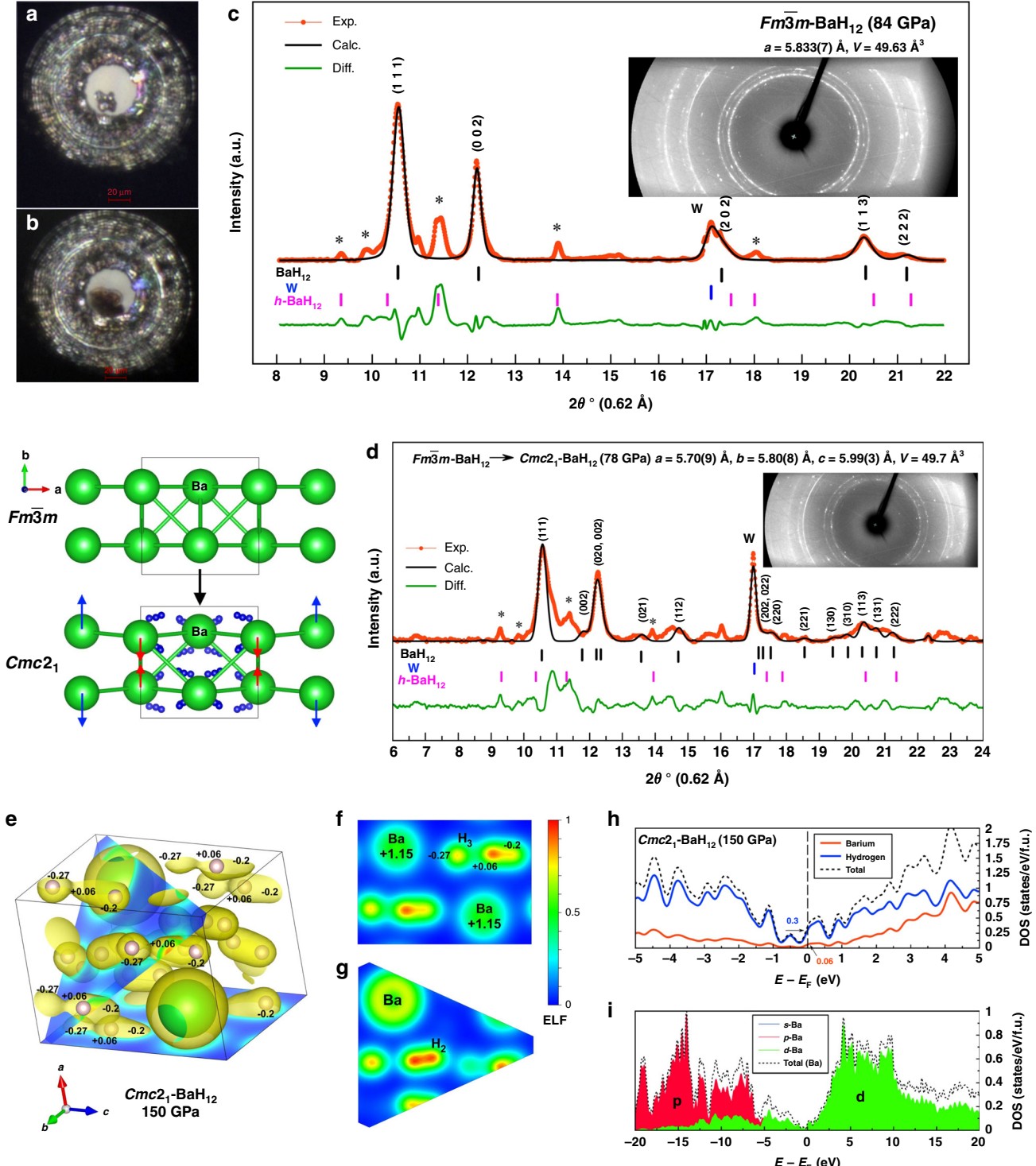

**Fig. 4 Low-pressure synthesis and physical properties of BaH₁₂.** Loaded Ba/AB sample **a** before and **b** after the laser heating in DAC #B3. Experimental XRD pattern and the Le Bail refinement of **c** the ideal $Fm\bar{3}m$-BaH₁₂ structure at 84 GPa and **d** $Cmc2_1$-BaH₁₂ obtained via the distortion of $Fm\bar{3}m$-BaH₁₂. The reflections indicated by asterisks may correspond to unidentified hexagonal barium polyhydride BaH₋₁₂. The experimental data, model fit for the structure, and residues are shown in red, black, and green, respectively. **e** Electron localization function (ELF), projected onto **f** the (100) plane and **g** the (11$\bar{1}$) plane, and Bader charges of the Ba and H atoms in $Cmc2_1$-BaH₁₂ at 150 GPa. **h** Contribution of barium and hydrogen to the electronic density of states of BaH₁₂. **i** d-character of Ba electrons near the Fermi level.

120-135 GPa the superconducting parameters of $I4/mmm$-BaH₁₂ and $Fm\bar{3}m$-BaH₁₂, isostructural to YB₁₂, the structures that were considered as possible solutions at the first step of the XRD interpretation. The calculations show that filling of the pseudogap in $N(E)$ makes it possible to reach $T_C \sim 200$ K with $\lambda \geq 3$ in these compounds (Supplementary Figs. S42 and S43).

In conclusion, studying the high-pressure chemistry of the Ba–H system in four independent DACs we successfully

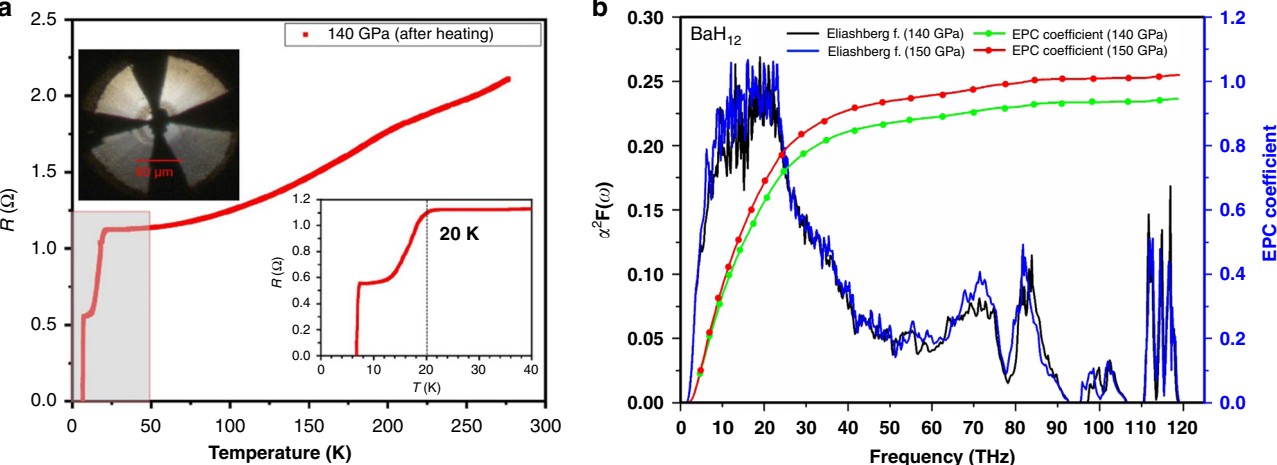

**Fig. 5 Electrical measurements and calculations of superconducting properties. a** Four-probe measurements of the electrical resistance $R(T)$ of the BaH$_x$ sample synthesized using the laser heating at 140 GPa. The superconducting transition was detected at ~20 K. **b** Calculated Eliashberg spectral functions and electron-phonon coupling parameter ($\lambda$) for pseudocubic BaH$_{12}$ at 140 (black) and 150 GPa (blue).

synthesized novel barium superhydride BaH$_{12}$ with a pseudocubic crystal structure, stabilized in the pressure range of 75–173 GPa. The compound was obtained by laser-heating metallic barium with an excess of ammonia borane compressed to 173, 160, 146, and 90 GPa. The Ba sublattice structure of BaH$_{12}$ was resolved using the synchrotron XRD, evolutionary structure prediction, and several postprocessing Python scripts, including an XRD matching algorithm. Discovered BaH$_{12}$ has unique metallic conductivity, localized in the layers of molecular hydrogen, and the highest hydrogen content (>92 mol%) among all metal hydrides synthesized so far. The experimentally established lower limit of stability of barium dodecahydride is 75 GPa. The third-order Birch–Murnaghan equation of state and unit cell parameters of BaH$_{12}$ were found in the pressure range of 75–173 GPa: $V_{100} = 45.47 \pm 0.13$ Å$^3$, $K_{100} = 305 \pm 8.5$ GPa, and $K'_{100} = 3.8 \pm 0.48$. The ab initio calculations confirm a small distortion of the ideal fcc-barium sublattice to space group $Cmc2_1$ or $P2_1$, determined by the presence of additional weak reflections in the diffraction patterns. The impurity phase analysis indicates possible presence of BaH$_6$ and BaH$_{10}$. According to the theoretical calculations and experimental measurements, BaH$_{12}$ exhibits metallic and superconducting properties, with $T_C = 20$ K at 140 GPa, and its crystal structure contains H$_2$ and H$_3^-$ groups. The results of these experiments confirm that the comparative stability of superhydrides increases with the increase of the period number of a hydride-forming element in the periodic table[20]. Our work opens prospects for the synthesis of even more hydrogen rich compounds like predicted LaH$_{16}$[34] and ErH$_{15}$[20], and new ternary high-$T_C$ polyhydrides in such systems as Ba-Y-H and Ba-La-H.

## Methods

**Experimental details**. The barium metal samples with a purity of 99.99% were purchased from Alfa Aesar. All diamond anvil cells (100 μm and 50 μm culets) were loaded with a metallic Ba sample and sublimated ammonia borane (AB) in an argon glove box. The tungsten gasket had a thickness of $20 \pm 2$ μm. The heating was carried out by 2–3 pulses of an infrared laser (1.07 μm, Nd:YAG), each pulse had a duration of 0.3–0.5 s. The temperature was determined using the decay of black-body radiation within the Planck formula. The applied pressure was measured by the edge of diamond Raman signal[35] using the Horiba LabRAM HR800 Ev spectrometer with an exposure time of 10 s. The XRD patterns from samples in diamond anvil cells (DACs) were recorded on the BL15U1 synchrotron beamline at the Shanghai Synchrotron Research Facility (SSRF, China) using a focused ($5 \times 12$ μm) monochromatic X-ray beam with a linear polarization (20 keV, 0.6199 Å). Mar165 CCD was used as a detector.

The experiment with DAC #B0 was carried out at the Advanced Photon Source, Argonne, U.S. The loaded particle was successively heated up to 2100 K using

millisecond-long pulses ($4 \times 0.04$ s) of a 1064 nm Yb-doped fiber laser. We used this pulsed laser heating mode to avoid the premature breakage of a diamond. The synchrotron XRD measurements (the X-ray wavelength was 0.2952 Å) were performed at the GSECARS of the Advanced Photon Source[36] with about $3 \times 4$ μm X-ray beam spot.

The experimental XRD images were analyzed and integrated using Dioptas software package (version 0.5)[37]. The full profile analysis of the diffraction patterns and the calculation of the unit cell parameters were performed in the Materials studio[38] and JANA2006[39] using the Le Bail method[40].

To investigate the electrical resistivity of barium polyhydrides, we performed 5 runs of measurements in Cu-Be DACs #E1-5 using the four-probe technique. The preparation of all cells was similar. Tungsten gasket with initial thickness of 250 μm was precompressed to about 25 GPa. Then a hole with a diameter of 20% bigger than the culet diameter was drilled using pulse laser (532 nm). Cubic boron nitride (c-BN) powder mixed with epoxy was used as an insulating layer. We filled the chamber with MgO and compressed it to about 5 GPa. Then, in the obtained transparent MgO layer, a hole with diameter about 40μm was drilled by laser. UV lithography was used to prepare four electrodes on the diamond culet. We deposited the 500 nm thick Mo layer by magnetron sputtering (field of 200 V at 300 K) and removed the excess of metal by acid etching. Four deposited Mo electrodes were extended by platinum foil. The chamber was filled with the sublimated ammonia borane (AB) and a small piece of Ba was placed on the culet of upper diamond with the four electrodes. All preparations were made in an argon glove box (O$_2$ < 0.1 ppm, H$_2$O < 0.01 ppm). After that, the DACs were closed and compressed to a required pressure. We used 1.07 μm infrared pulse (~0.1 s, 1600 K) laser to heat the Ba/AB samples. Electrical resistance of the samples was studied in a cryostat (1.5-300 K, JANIS Research Company Inc.; in magnetic fields 0-9 T, Cryomagnetics Inc.) with applied current of 1 mA. More details about DACs #E1-5 are given in the Table S20.

**Computational details**. The study is based on the structural search for stable compounds in the Ba–H system using the USPEX code, for pressures of 50, 100, 150, 200, and 300 GPa, with a variable-composition evolutionary search from 0 to 24 atoms of each type (Ba, H). The first generation of the search (120 structures) was created using a random symmetric generator, all subsequent generations (100 structures) contained 20% of random structures and 80% of those created using the heredity, soft mutation, and transmutation operators. The results contain the files extended_convex_hull and extended_convex_hull_POSCARS, which were postprocessed using the Python scripts change_pressure.py, split_CIFs.py and xr_screening.py (see Scripts for XRD Postprocessing with USPEX section). The postprocessing script change_pressure.py performs an isotropic deformation of the unit cell of structures predicted by USPEX, bringing them to approximately experimental pressure. All three lattice constants of the structures are multiplied by a factor k, calculated under the assumption of validity of the Birch–Murnaghan equation of state[28] with the bulk modulus $K_0 = 300$ GPa and its first derivative $K' = 3$. This approach is a quick alternative to the script that uses a crude DFT reoptimization of a set of theoretically possible structures, bringing them to the experimental pressure. The script split_CIFs.py converts the set of POSCARS recorded in the extended_convex_hull_POSCARS file into a set of CIF files, simultaneously symmetrizing the unit cells and sorting the files by ascending fitness (the distance from the convex hull). The CIF files created in such a way can be directly analyzed using Dioptas[37], JANA2006[39] and other software. Finally, the script xr_screening.py automatically searches for the structures found by USPEX

and translated to the experimental pressure that exhibit a high similarity between the experimental and predicted XRD patterns (the latter are obtained using pymatgen[41] Python library). The analysis of complex mixtures consisted of two steps: first, we searched for the main component having the most intense reflections, then the already explained reflections were excluded to analyze the side phases.

To calculate the equations of state (EoS) of $BaH_{12}$, we performed structure relaxations of phases at various pressures using the density functional theory (DFT)[42,43] within the generalized gradient approximation (Perdew–Burke–Ernzerhof functional)[44] and the projector augmented wave method[45–49] as implemented in the VASP code[15–17]. The plane wave kinetic energy cutoff was set to 1000 eV, and the Brillouin zone was sampled using the Γ-centered k-points meshes with a resolution of $2\pi \times 0.05$ Å$^{-1}$. The obtained dependences of the unit cell volume on the pressure were fitted by the Birch–Murnaghan equation[28] to determine the main parameters of the EoS — the equilibrium volume $V_0$, bulk modulus $K_0$, and its derivative with respect to pressure K' — using EOSfit7 software[50]. We also calculated the phonon densities of states for the studied materials using the finite displacement method (VASP and PHONOPY)[51,52].

The calculations of the phonons, electron-phonon coupling, and superconducting $T_C$ were carried out with QUANTUM ESPRESSO (QE) package[53,54] using the density functional perturbation theory[55], employing the plane-wave generalized gradient approximation with Perdew–Burke–Ernzerhof functional[44]. In our ab initio calculations of the electron-phonon coupling (EPC) parameter λ of $Cmc2_1$-$Ba_4H_{48}$, the first Brillouin zone was sampled by $2 \times 2 \times 2$ q-points mesh and $4 \times 4 \times 4$ or $8 \times 8 \times 8$ k-points meshes with the smearing σ = 0.005–0.05 Ry that approximates the zero-width limits in the calculation of λ. The critical temperature $T_C$ was calculated using the Allen–Dynes equations[56].

Bader charges were calculated using Critic2[57,58] software with the atomic partition generated using the YT[59] method. The electron localization function (ELF) of $BaH_{12}$ and isosurface are shown for an isovalue of 0.12. The lattice planes (100) and (11$\bar{1}$) are shown at distances from the origin of 0 and 2.289 Å, respectively. We projected the ELF on these planes to show $H_3$ and $H_2$ bonding.

The average structure of $BaH_{12}$ was analyzed using the ab initio molecular dynamics (AIMD) simulations within the general gradient approximation[44] and using the augmented plane wave method[45,47–49] implemented in VASP software[30–32]. The total number of atoms in the model was 52, including 48 hydrogen atoms and 4 barium atoms ($Cmc2_1$-$Ba_4H_{48}$). The positions of the barium atoms were fixed during the simulation. The energy cutoff was set to 400 eV. The behavior of the hydrogen atoms in the $BaH_{12}$ crystal structure was studied upon annealing from 1500 to 10 K using the Nosé–Hoover thermostat[60,61]. The total simulation time was 10 ps with the time step of 1 fs.

## Data availability
The authors declare that the data supporting the findings of this study are available within the paper and its supplementary information files.

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

## Acknowledgements

The authors thank the staff of the Shanghai Synchrotron Radiation Facility and express their gratitude to Bingbing Liu's group (Jilin University) for their help in the laser heating of samples. This work was supported by the National Key R&D Program of China (Grant No. 2018YFA0305900), the National Natural Science Foundation of China (Grant Nos. 51632002, 11974133 and 51720105007), the Program for Changjiang Scholars and Innovative Research Team in University (Grant No. IRT_15R23). Work of D.V.S. and A.G.K. is supported by Russian Science Foundation (Grant No. 19-72-30043); work of A. R.O. is funded by Russian Ministry of Science and Technology (Grant 2711.2020.2 to leading scientific schools). Portions of this work were performed at GeoSoilEnviroCARS (The University of Chicago, Sector 13), Advanced Photon Source (APS), Argonne National Laboratory. GeoSoilEnviroCARS is supported by the National Science Foundation — Earth Sciences (EAR — 1634415) and Department of Energy — GeoSciences (DE-FG02-94ER14466). This research used resources of the Advanced Photon Source, a U.S. Department of Energy (DOE) Office of Science User Facility operated for the DOE Office of Science by Argonne National Laboratory under Contract No. DE-AC02-06CH11357. A.G.K., D.V.S. thank the Russian Foundation for Basic Research (Grant No. 19-03-00100). The reported study was funded by RFBR, project 20-32-90099. We thank the Russian Ministry of Science and Higher Education agreement No. 075-15-2020-808. A.F.G. acknowledges the support of the Army Research Office. We thank Dr. Christian Tantardini (Skoltech) for help with the calculation of the Bader charges, Dr. Di Zhou (JLU) for preparing the convex hulls and $\alpha^2 F(\omega)$ diagrams, and Igor Grishin (Skoltech) for proofreading of the manuscript.

## Author contributions

X.H., W.C., D.V.S., A.G.K., A.R.O. and T.C. conceived this project. W.C., D.V.S., X.H. performed the experiment, D.V.S., A.G.K, I.A.K., H.S., D.D., A.R.O., M.G. and T.C. prepared the theoretical calculations and analysis. W.C., X.H., A.F.G. and V.B.P. performed X-ray measurements of synthesized samples. W.C., D.V.S., A.G.K., A.R.O., X.H. and T.C. wrote and revised the paper. All the authors discussed the results and offered useful inputs.

## Competing interests

The authors declare no competing interests.
