## [Peer Review File · Nature Communications]

REVIEWER COMMENTS

Reviewer #1 (Remarks to the Author):

The manuscript reports on the first experimental synthesis and detection of metallic superconducting Ba hydride under high pressure. This discovery is very important because it can shed light on the mechanisms behind the chemistry and formation of high temperature superhydrides, and introduces a newly experimentally verified family of binary hydrides in the periodic table.

In spite of the interest and the importance of the findings reported in the paper, I found several flaws which prevent the manuscript to be published in the present status.

These are the major points:

- the authors should show a rigorous comparison of the X-ray diffraction pattern between the most stable BaH₁₂ symmetries (Cmc2_1, P2_1, P1) and experimental data. Only the Cmc2_1 pattern is shown in the manuscript.
- Why Cmc2_1 is the one detected experimentally even if the theoretical calculations clearly predict lowers symmetry structures as the most stable? Anharmonicity together with the nuclear quantum effects (NQE) are the most likely reasons. The authors should comment on that in the paper. NQE with anharmonic potentials can stabilize different symmetries than the ones obtained by a harmonic approach.
- The effect of impurities in the sample are not well discussed. According to Fig.3 the impurities are attributed to BaH₁₂ stoichiometry, while in the abstract and in the conclusions BaH₁₀ and BaH₆ are cited as impurities, based on the convex hull diagram in Fig.1(c). This is confusing, and must be clarified.
- The synthesis of the "B" samples/cells is very well detailed, while for the "E" cells no description is reported. Please, add more information also for the "E" cells.
- In the theoretical analysis of superconductivity, which are the phonon modes that carry the largest el-ph coupling? In Fig. 5(b), reporting the physical values for different smearings is useless. One can systematically check the convergence as a function of smearing, at a given q-mesh. Only the converged results are of interest. Moreover, the 39-53 K range is reported in the text for the superconducting critical temperature at 150 GPa. The last sentence of the paragraph is not clear: "... and at 150 GPa the expected T_c may reach 59K", which is out of range. The authors should report in the main text at which pressure I4/mmm and Fm-3m yield the theoretically predicted superconducting T_c.

As a general comment, I found that the results in the paper are presented in a quite fragmented way. An effort to make them more homogeneously and uniformly organized must be done by the authors.

Other points:

In the Supporting Information, it is explained that the electron-phonon coupling calculations are done with LDA, while all other calculations in PBE. Why the functional has been changed? The phonon and electron-phonon coupling calculations must be done at the relaxed structures. It is possible that PBE and LDA give different equilibrium structures and so different electron-phonon couplings and T_c. Please, verify.

In the S1 convex hull diagrams, one would expect to find not only the Cmc2_1 symmetry, but also the P2_1 and P1. Moreover, a direct comparison the the convex hull diagrams with and without ZPE would

be useful, to study the impact of NQE at least in the harmonic approximation.

Page S26. Please, verify the cross references.

Reviewer #2 (Remarks to the Author):

In the manuscript entitled "Synthesis of Molecular Metallic Barium Superhydride: Pseudocubic BaH₁₂", Wuhao Chen et al. present a combined experimental and theoretical study to reveal the formation of a new superhydride BaH₁₂ structure at pressures ranging from 75 to 173 GPa. In contrast with recently synthesized high-T_c superhydrides such as LaH₁₀, YH₆, ThH₁₀, CeH₉, and so on, BaH₁₂ contains the molecular units of H₂ and H₃ as well as detached H₁₂ chains. Interestingly, they argue that the latter one-dimensional chains are formed as a result of the Peierls-type distortion of a cage-like structure. Moreover, this superhydride is observed to exhibit a superconducting transition of ~20 K at 140 GPa, which is much lower than those of other superhydrides with cage structures.

Unfortunately, I do not see sufficient novelty and significance which can warrant publication of these results in Nature Communications and I cannot find this work to be interesting for a general audience of Nature Communications. In this referee's opinion, the current work is well suited for a more specialized journal. But I find a few deficiencies in the paper which are listed below:

1. The authors claimed that "The computed equation of state of Fm-3m BaH₁₂ (Fig.1d) corresponds well to the experimental volume-pressure dependence." In Fig. 1d, the computed equation of states of Fm-3m BaH₁₂ shows a phase transition, but it seems that there is no phase transition in the experiment results. I cannot find a good agreement between theory and experiment. What is the difference between the phases before and after the phase transition? How about this kind of phase transition in Cmc21 or P21 phase?
2. The authors claimed that the semimetallic Cmc21 phase of BaH₁₂ explains well the experimental results. However, it is dynamically unstable (see Fig. 2). How does this unstable structure correspond to the experimental structure?
3. How do you calculate the superconducting T_c of the pseudocubic Cmc21-BaH₁₂, which has the imaginary phonon modes.
4. The observed T_c values show a large difference between 20 and 7 K at 140 GPa and 132 GPa, respectively. Why does such large difference of T_c occur?
5. How do the predicted T_c values of Cmc21-BaH₁₂ vary as a function of pressure? Do they agree well with the experimental data?
6. The authors claimed that "The low electronic density of state in semimetallic Cmc21 BaH₁₂ looks typical for one-dimensional ...H-H-H... chains, which are divided into H₂, H₃ fragments due to the Peierls-type distortion. In fact, all of the discussed structures of BaH₁₂ can be viewed as a result of Peierls-type distortion." I cannot find the proper origin for this Peierls-distortion. Can you provide more detailed explanation for the Peierls-distortion mechanism?

Reviewer #1 (Remarks to the Author):

The manuscript reports on the first experimental synthesis and detection of metallic superconducting Ba hydride under high pressure. This discovery is very important because it can shed light on the mechanisms behind the chemistry and formation of high temperature superhydrides, and introduces a newly experimentally verified family of binary hydrides in the periodic table.

In spite of the interest and the importance of the findings reported in the paper, I found several flaws which prevent the manuscript to be published in the present status.

These are the major points:

- the authors should show a rigorous comparison of the X-ray diffraction pattern between the most stable BaH₁₂ symmetries (*Cmc2*₁, *P2*₁, *P1*) and experimental data. Only the *Cmc2*₁ pattern is shown in the manuscript.

REPLY: We thank the reviewer for his interest to this research. This comparison was added to the Supporting Information. We also added discussion of this comparison in the main text. In fact, all these structures differ only in the structure of hydrogen sublattice, while the X-ray diffraction patterns come from the sublattice of barium atoms. Thus, XRD patterns of these structures are similar (pseudocubic) and practically cannot be distinguished from each other.

Fig. R1. Comparison of the predicted XRD patterns of DFT relaxed (PAW_PBE) structures of *Cmc2*₁, *P2*₁ and *P1*-BaH₁₂ at 150 GPa (before refinement). All considered structures have a pseudocubic diffraction pattern and a slight distortion of these structures in experimental conditions can transform the patterns to each other.

- Why *Cmc2*₁ is the one detected experimentally even if the theoretical calculations clearly predict lower symmetry structures as the most stable? Anharmonicity together with the nuclear quantum effects (NQE) are the most likely reasons. The authors should comment on that in the paper. NQE with anharmonic potentials can stabilize different symmetries than the ones obtained by a harmonic approach.

REPLY: The $Cmc2_1$ modification was chosen for the best fit with the observed experimental properties - metallic conductivity, superconductivity and absence of Raman signals. It is possible that the correct structural solution for the hydrogen sublattice is a combination of the found H-nets in the $Cmc2_1$ -, $P2_1$ - and $P1$ -, and can only be found within supercells size > 100 atoms. We believe that such complex calculations, together with a detailed analysis of the impact of anharmonic effects on the structure, electronic properties and superconductivity of the compound, should be the subject of a separate work.

At the moment, there is no experimental technique to establish the exact structure of the hydrogen sublattice at pressures above 100 GPa. Thus, the solution found theoretically cannot be strictly justified experimentally, even for the famous H_3S and LaH_{10} superconductors.

We have studied the influence of anharmonicity in the framework of molecular dynamics. It confirms that the most stable modification at 300 K is also a pseudocubic $P1$ - BaH_{12} . However, as already was mentioned, the most thermodynamically stable phase is semiconducting, whereas in all experiments our samples were good metals even at pressures below 100 GPa.

- The effect of impurities in the sample are not well discussed. According to Fig.3 the impurities are attributed to BaH_{12} stoichiometry, while in the abstract and in the conclusions BaH_{10} and BaH_6 are cited as impurities, based on the convex hull diagram in Fig.1(c). This is confusing, and must be clarified.

REPLY: discussion of these impurities has been moved to the Supporting Information, section "Synthesis at 173 and 154 GPa: Side Products". In fact, the XRD reflections of potentially existing BaH_{10} and BaH_6 overlap with the BaH_{12} reflections and, therefore, we have insufficient information for identification of the impurities. BaH_6 and BaH_{10} are proposed as possible candidates on the basis of theoretical modeling, simple indexing of the reflections in hexagonal space group (i.e. 'h- BaH_{12} ') and calculations of the unit cell volumes at corresponding pressures. In Fig. 3 intensity of side reflections is not enough for a confident interpretation, but the peaks marked as (*) can be indexed in $P63/mmc$ space group and corresponding unit cell volume is close to volume of BaH_{12} .

- The synthesis of the "B" samples/cells is very well detailed, while for the "E" cells no description is reported. Please, add more information also for the "E" cells.

REPLY: Description of the "E"-series of diamond anvil cells was added to the Supporting Information ("Methods").

- In the theoretical analysis of superconductivity, which are the phonon modes that carry the largest el-ph coupling? In Fig. 5(b), reporting the physical values for different smearings is useless. One can systematically check the convergence as a function of smearing, at a given q-mesh. Only the converged results are of interest. Moreover, the 39-53 K range is reported in the text for the superconducting critical temperature at 150 GPa. The last sentence of the paragraph is not clear: "... and at 150 GPa the expected T_c may reach 59K", which is out of range. The authors should report in the main text at which pressure $I4/mmm$ and $Fm-3m$ yield the theoretically predicted superconducting T_c .

REPLY: The phonon modes corresponding to the largest el-ph coupling are low-frequency soft modes associated with both barium and hydrogen atoms. We have changed Fig. 5 (b) in the proposed manner. "59 K" – is a misprint, should be 53 K. Pressures (120-135 GPa) for which we calculated T_c of $I4/mmm$ and $Fm-3m$ were added.

As a general comment, I found that the results in the paper are presented in a quite fragmented way. An effort to make them more homogeneously and uniformly organized must be done by the authors.

REPLY: in the revised version the text was rewritten and its coherence improved. Newly added material is highlighted in blue.

Other points:

In the Supporting Information, it is explained that the electron-phonon coupling calculations are done with LDA, while all other calculations in PBE. Why the functional has been changed? The phonon and electron-phonon coupling calculations must be done at the relaxed structures. It is possible that PBE and LDA give different equilibrium structures and so different electron-phonon couplings and T_c . Please, verify.

REPLY: We are sorry for misleading, this was a misprint. All calculations were performed within the same functional (PBE).

In the S1 convex hull diagrams, one would expect to find not only the Cmc2_1 symmetry, but also the P2_1 and P1. Moreover, a direct comparison the convex hull diagrams with and without ZPE would be useful, to study the impact of NQE at least in the harmonic approximation.

REPLY: the direct comparison of Ba-H convex hulls with and without ZPE at 0 K was added to the Supporting Information (Fig. S4). At 150 GPa ZPE stabilizes Immm-BaH12, however, this compound has not been detected. At 100 GPa and below, Cmc21-BaH12 is close to the convex hull without accounting of ZPE. Discrepancies between the experiment and results of theoretical calculations may point to an insufficient accuracy of the known pseudopotentials of Ba and H in the studied range of pressures.

Page S26. Please, verify the cross references.

REPLY: We fixed this issue.

Reviewer #2 (Remarks to the Author):

In the manuscript entitled “Synthesis of Molecular Metallic Barium Superhydride: Pseudocubic BaH12”, Wuhaio Chen et al. present a combined experimental and theoretical study to reveal the formation of a new superhydride BaH12 structure at pressures ranging from 75 to 173 GPa. In contrast with recently synthesized high- T_c superhydrides such as LaH10, YH6, ThH10, CeH9, and so on, BaH12 contains the molecular units of H2 and H3 as well as detached H12 chains. Interestingly, they argue that the latter one-dimensional chains are formed as a result of the Peierls-type distortion of a cage-like structure. Moreover, this superhydride is observed to exhibit a superconducting transition of ~ 20 K at 140 GPa, which is much lower than those of other superhydrides with cage structures.

Unfortunately, I do not see sufficient novelty and significance which can warrant publication of these results in Nature Communications and I cannot find this work to be interesting for a general audience of Nature Communications. In this referee’s opinion, the current work is well suited for a more specialized journal.

REPLY: BaH12 is a unique novel superhydride which plays a role of connecting bridge between two large classes of superhydrides:

1) molecular hydrides of alkali metals with semiconducting or dielectric properties, e.g. CaH4 (*J. Chem. Phys* 2019), LiH6 (*PNAS* 2015), NaH7 (*NatComm* 2016), CsH7 (*PRB* 2020)..., and

2) high- T_c superconducting metal hydrides like LaH10 (*Nature* 2019), YH6 and YH9, ThH10 (*MatToday* 2020), CeH9..., etc. in which hydrogen forms a high-symmetric clathrate structure.

BaH12 combines key features of both types of superhydrides: molecular structure with metallic properties and superconductivity. In the same time, the discovered here barium superhydride has one of the highest hydrogen content of all known hydrides. With 12 atoms of hydrogen per each Ba it is stable even below 100 GPa. Our work opens prospects for the synthesis of even more hydrogen rich compounds, such as predicted LaH16 and ErH15, and new ternary high- T_c polyhydrides in such systems as Ba-Y-H, Ba-La-H.

But I find a few deficiencies in the paper which are listed below:

1. The authors claimed that “The computed equation of state of Fm-3m BaH₁₂ (Fig.1d) corresponds well to the experimental volume-pressure dependence.” In Fig. 1d, the computed equation of states of Fm-3m BaH₁₂ shows a phase transition, but it seems that there is no phase transition in the experiment results. I cannot find a good agreement between theory and experiment. What is the difference between the phases before and after the phase transition? How about this kind of phase transition in Cmc21 or P21 phase?

REPLY: The equation of state of the Fm-3m-BaH₁₂ was compared to the experiment only above 100 GPa where the agreement is satisfactory. The phase transition below 100 GPa is irrelevant as we ruled out Fm-3m-BaH₁₂ because of its thermodynamic instability. Possible phase transitions in low-symmetry modifications of BaH₁₂ were also studied. There should be $P2_1$ -BaH₁₂ \rightarrow $Immm$ -BaH₁₂ transition at \sim 190 GPa (Fig. R2). However, this pressure is beyond the experimentally studied region and $Immm$ -BaH₁₂ was not detected experimentally.

Fig. R2. Enthalpy-pressure diagram for low-symmetry modifications of BaH₁₂ at 100-300 GPa, 0 K.

2. The authors claimed that the semimetallic Cmc21 phase of BaH₁₂ explains well the experimental results. However, it is dynamically unstable (see Fig. 2). How does this unstable structure correspond to the experimental structure?

REPLY: XRD pattern of $Cmc2_1$ -BaH₁₂ best fits to experimental X-ray diffraction. Electronic properties of $Cmc2_1$ -BaH₁₂ correspond well to observed metallic and superconducting properties of the synthesized barium superhydride. As we demonstrated, within DFT approach the $Cmc2_1$ -BaH₁₂ should undergo a distortion to $P1$. However, this distortion is not observed experimentally. Additional factors related to the questions, which are beyond the scope of this work: 1) anharmonic effects, 2) limitations of available Ba pseudopotentials.

3. How do you calculate the superconducting T_c of the pseudocubic Cmc21-BaH₁₂, which has the imaginary phonon modes.

REPLY: Imaginary phonon modes were excluded from the calculations of isotropic Eliashberg spectral function performed via averaging of $\omega_{q\nu}\lambda_{q\nu}$ over the BZ.

4. The observed T_c values show a large difference between 20 and 7 K at 140 GPa and 132 GPa, respectively. Why does such large difference of T_c occur?

REPLY: Because BaH₁₂ is near the metallization pressure (Fig. 2c, Fig. 4h). Even a small change in pressure in this region (120-140 GPa) leads to a relatively large change in the density of electronic states at the Fermi level and, therefore, in the superconducting properties.

5. How do the predicted T_c values of Cmc21-BaH₁₂ vary as a function of pressure? Do they agree well with the experimental data?

REPLY: We have calculated T_c at 2 points of pressure: 140 and 150 GPa (see revised Fig. 5), and results point to a substantial increase of the critical temperature along with the pressure (dT_c/dP = +0.7 K). The available experimental data confirm this trend: we observed superconducting transition at about 5–7 K (132 GPa) and ~20 K at 140 GPa.

Theoretically predicted T_c of BaH₁₂ is higher than experimental values, but this is well-known fact for superhydrides. Typically, DFT calculations within harmonic approach overestimate T_c, for instance: LaH₁₀ (predicted: 286 K, found: 250 K), YH₆ (predicted: >270 K, found: 224 K), YH₉ (predicted: 303 K, found: 243 K), ThH₁₀ (predicted: 221 K, found 160 K) etc. Thus, the observed discrepancy (~20-30 K) between theoretical and experimental T_c is expected.

6. The authors claimed that “The low electronic density of state in semimetallic Cmc21 BaH₁₂ looks typical for one-dimensional ...H-H-H... chains, which are divided into H₂, H₃ fragments due to the Peierls-type distortion. In fact, all of the discussed structures of BaH₁₂ can be viewed as a result of Peierls-type distortion.” I cannot find the proper origin for this Peierls-distortion. Can you provide more detailed explanation for the Peierls-distortion mechanism?

REPLY: 1D chain of hydrogen atoms provides a classic example of Peierls distortion. The result of the distortion is changing distances between atoms in the chain. This leads to consequent decomposition of the chain into H₂ molecules with formation of stable non-metallic molecular hydrogen from unstable metallic modification. The Peierls effect is opposed by repulsive nearest neighbor forces that can keep the structure from distortion. Under high pressure, these forces become more pronounced, suppressing Peierls distortion and stabilizing the metallic state.

REVIEWER COMMENTS

Reviewer #1 (Remarks to the Author):

The authors improved their manuscript. However, there are still issues that prevent me to recommend publication in the present form.

The first and more serious flaw is the inconsistency between the main text and the reply to Reviewer #1 about the difference of the X-ray pattern between the putative distorted pseudocubic phases. In the reply the authors state that:

"Thus, XRD patterns of these structures are similar (pseudocubic) and practically cannot be distinguished from each other"

In the main text (line 18 page 4) it is stated that:

"The comparative analysis of Cmc21, P21, and P1 structures of BaH12 shows that semimetallic Cmc21 best explains the experimental results of the X-ray diffraction,...",
and later:

"P21-BaH12 shows a complex picture of splitting of the diffraction signals,"

From the added Fig. S2, it is clear that it is not possible to choose one of the three phases as *best* candidate based on the X-ray diffraction pattern. From the same Fig. S2, I also do not see that P21 should have a more complex picture of peak splitting than the other two structures. P1, and not P21, is certainly the most complex. Thus, the sentence in the main text has to be corrected.

Always about line 18 page 4. "Cmc21.....lies much closer to the convex hull than F3-m3 and I4/mmm modifications". The F3-m3 and I4/mmm points have never been shown in the convex hull diagram. The new Figure S4 (or Tables S7-S12) are the ideal place to show also the energetics of F3-m3 and I4/mmm symmetries, and give a more quantitative meaning to the sentence.

It is also written that "the equation of state (of P21-BaH12) deviates from the experimental data". Also in this case, the EOS for P21-BaH12 is not shown. It would be interesting and useful, indeed, to compare the EOS for the 3 phases (Cmc21, P21, and P1) against the experimental one, but this comparison is not shown at all. Only Cmc21 is reported in Fig. 1(d).

A side remark on Fig. S39. Please, report only the converged results of the DOS. It is useless to show unconverged smearings! The tetrahedron (possibly on a denser energy grid) is enough.

The authors made an effort to better integrate the different parts of the paper, and they tried to improve its coherence. However, a better referencing of the supporting information should be done in the main text. For instance, new Figs. S1, S4 and S5 are not referenced in the paper. For the supplemental material which is cited, there are wrong references, like for instance the following one (line 89 Page 3):

"The analysis of the experimental data within space group F3-m3 (Fig. 1b and Supporting Table S5)..."

This should be Table S3, instead of Tab. S5.

This is not acceptable for a high quality publication and must be amended.

Reviewer #2 (Remarks to the Author):

I think that the authors have appropriately addressed most of comments raised by the reviewer. However, there are still some minor issues to be resolved before publication in Nature Comm., as follows:

(1) In the response to Comment 1, the authors claimed that there should be a phase transition from P21 to Immm above 190 GPa (Fig. R2). However, only P21, P1, and Immm phases are considered in

Fig. R2. How about the enthalpy-pressure diagram for the Cmc21 phase?

(2) Regarding Comment 2, the authors explained the stability of the P1 structure in terms of anharmonic effects and Ba pseudopotentials. However, I don't understand how these reasons influence the energetics between the Cmc21 and P1 phases.

Reviewer #1 (Remarks to the Author):

The authors improved their manuscript. However, there are still issues that prevent me to recommend publication in the present form.

The first and more serious flaw is the inconsistency between the main text and the reply to Reviewer #1 about the difference of the X-ray pattern between the putative distorted pseudocubic phases. In the reply the authors state that:

"Thus, XRD patterns of these structures are similar (pseudocubic) and practically cannot be distinguished from each other". In the main text (line 18 page 4) it is stated that: "The comparative analysis of Cmc21, P21, and P1 structures of BaH12 shows that semimetallic Cmc21 best explains the experimental results of the X-ray diffraction,...", and later: "P21-BaH12 shows a complex picture of splitting of the diffraction signals,"

From the added Fig. S2, it is clear that it is not possible to choose one of the three phases as *best* candidate based on the X-ray diffraction pattern. From the same Fig. S2, I also do not see that P21 should have a more complex picture of peak splitting than the other two structures. P1, and not P21, is certainly the most complex. Thus, the sentence in the main text has to be corrected.

REPLY: we apologize for the inaccuracies we have made. Misprints (like "P21" instead of "P1") and mistakes specified by the reviewer were corrected in the new version of manuscript.

Always about line 18 page 4. "Cmc21..... ..lies much closer to the convex hull than F3-m3 and I4/mmm modifications". The F3-m3 and I4/mmm points have never been shown in the convex hull diagram. The new Figure S4 (or Tables S7-S12) are the ideal place to show also the energetics of F3-m3 and I4/mmm symmetries, and give a more quantitative meaning to the sentence.

REPLY: We have added the information about the stability of Fm-3m and I4/mmm modifications to the Fig S4, to the Tables S7-S12 and to the main text. The calculated enthalpy of the Fm-3m-BaH₁₂ turns out to be positive and goes beyond the convex hulls, which indicates the potential instability of this structure. Phonon densities of state of I4/mmm-BaH₁₂ at 135-170 GPa (Fig. S5b) point to a dynamical instability of this structure.

It is also written that "the equation of state (of P21-BaH12) deviates from the experimental data". Also in this case, the EOS for P21-BaH12 is not shown. It would be interesting and useful, indeed, to compare the EOS for the 3 phases (Cmc21, P21, and P1) against the experimental one, but this comparison is not shown at all. Only Cmc21 is reported in Fig. 1(d).

REPLY: We added new data on the EOS of Cmc21, P21, P1 to the Fig. S1b (Supporting Information, see below) showing the detailed EOS of Cmc21, P21, and P1 in comparison with experimental data.

A side remark on Fig. S39. Please, report only the converged results of the DOS. It is useless to show unconverged smearings! The tetrahedron (possibly on a denser energy grid) is enough.

REPLY: We replaced Fig. S38a (former Fig. S39) by the new one where the tetrahedron method was used for calculations using both PBE and LDA.

The authors made an effort to better integrate the different parts of the paper, and they tried to improved its coherence. However, a better referencing of the supporting information should be done in the main text. For instance, new Figs. S1, S4 and S5 are not referenced in the paper.

REPLY: We fixed this issue

For the supplemental material which is cited, there are wrong references, like for instance the following one (line 89 Page 3):

"The analysis of the experimental data within space group F3-m3 (Fig. 1b and Supporting Table S5)..."

This should be Table S3, instead of Tab. S5.

This is not acceptable for a high quality publication and must be amended.

REPLY: We thank referee for careful reading, we fixed these typos and double check all text.

Reviewer #2 (Remarks to the Author):

I think that the authors have appropriately addressed most of comments raised by the reviewer.

REPLY: We thank referee for appreciation of our revised version of the manuscript.

However, there are still some minor issues to be resolved before publication in Nature Comm., as follows:

(1) In the response to Comment 1, the authors claimed that there should be a phase transition from P21 to Immm above 190 GPa (Fig. R2). However, only P21, P1, and Immm phases are considered in Fig. R2. How about the enthalpy-pressure diagram for the Cmc21 phase?

REPLY: We added new data about the phase transition to the Enthalpy-pressure diagram in the Fig. S5a (Supp.Info.). Cmc21-BaH₁₂ has higher enthalpy compared to P21, P1, and Immm modifications. This information can be found from the convex hull calculated at different pressures shown in Fig. S4, where we added the information about Fm-3m and I4/mmm phases as well.

(2) Regarding Comment 2, the authors explained the stability of the P1 structure in terms of anharmonic effects and Ba pseudopotentials. However, I don't understand how these reasons influence the energetics between the Cmc21 and P1 phases.

REPLY: Chosen pseudopotential of Ba influences the energy of Ba-H interactions. The Ba-H bond geometry and total enthalpy differ for each of the considered BaH₁₂ modifications (Cmc2₁, P2₁, P1, Immm, I4/mmm) since the topology of the hydrogen sublattice is different. Thus, $\Delta H(\text{Cmc}2_1\text{-P}2_1)$ will depend on the selected barium pseudopotential. The anharmonic contribution to ZPE will also vary for different BaH₁₂ structures and will impact on $\Delta G(\text{Cmc}2_1\text{-P}2_1)$. As an illustration, we calculated ΔH for PBE and LDA sets of pseudopotentials (Table R1). One can see that $\Delta H(\text{Cmc}2_1\text{-P}2_1, \text{PBE}) = 55 \text{ meV/atom}$, while $\Delta H(\text{Cmc}2_1\text{-P}2_1, \text{LDA}) = 35 \text{ meV/atom}$.

Table R1. Calculated difference in the enthalpy of various BaH₁₂ modifications (135 GPa).

X=	H(eV/f.u.), PBE	H(eV/f.u.), LDA	$\Delta H(\text{X-Cmc}2_1)$, eV/atom PBE	$\Delta H(\text{X-Cmc}2_1)$, eV/atom LDA
I4/mmm-BaH ₁₂	10.5822	5.720308	0.016315385	0.008494898
Immm-BaH ₁₂	9.726517	5.164016	-0.049506385	-0.034296756
Cmc2 ₁ -BaH ₁₂	10.3701	5.609874	0	0
P2 ₁ -BaH ₁₂	9.654088	5.147783	-0.055077885	-0.035545482
P1-BaH ₁₂	9.747719	5.268068	-0.047875481	-0.026292767

REVIEWERS' COMMENTS

Reviewer #1 (Remarks to the Author):

The manuscript has considerably improved. The discussion of different possible polymorphic modifications of BaH12 at different pressures is now clear, and will certainly stimulate further studies. The outcome of the paper is significant and of potentially high impact, as I highlighted in my previous review. Therefore, I recommend its publication as is.

Reviewer #2 (Remarks to the Author):

The authors have now satisfactorily addressed my concerns and modified the manuscript and supplemental information file accordingly. I recommend the publication of this manuscript in its current form in Nature Communications.